

# Nonpharmacological pain relief for labour pain: knowledge, attitude, and barriers among obstetric care providers

Heba Abdel-Fatah Ibrahim[1], Majed Said Alshahrani[2], Amlak Jaber Al-Qinnah[3] and Wafaa Taha Elgzar[4]

[1] Department of Maternity and Childhood Nursing, Nursing College, Najran University, Najran, Saudi Arabia
[2] Department of Obstetrics and Gynecology, College of Medicine, Najran University, Najran, Saudi Arabia
[3] Maternity and Children Hospital, Najran, Saudi Arabia
[4] Department of Maternity and Childhood Nursing, Nursing College, Najran University, Najran, Saudi Arabia

Corresponding authors
Majed Said Alshahrani,
msalshahrane@nu.edu.sa
Wafaa Taha Elgzar,
wtelgzar@nu.edu.sa

## ABSTRACT

**Background**. Labor pain is considered the worst pain in a woman's life. Hence, pain control should be essential to labor management at any level. There is scarce information, and there are gaps regarding the knowledge, attitude, and barriers to the utilization of nonpharmacological approaches for pain relief in Saudi Arabia. Therefore, the current study aims to evaluate nonpharmacological pain relief (NPPR)-related knowledge, attitudes, and barriers among obstetric care providers in Najran, Saudi Arabia.

**Methods**. A cross-sectional analytical study was performed at maternity departments in Maternal and Children Hospital (MCH), Najran, Saudi Arabia, from April 1 to May 26 2023. The study involved 186 obstetric care providers (OPCs), physicians (19), nurses (144), and midwives (23). A structured self-reported questionnaire was used to collect data and involves five main sections: demographic data, work-related data, nonpharmacological pain relief-related attitude, perceived barriers, and knowledge quiz. The adjusted odds ratio (AOR) along with 95% CI was estimated to determine the factors associated with nonpharmacological pain relief-related knowledge and attitude using multivariate analysis in the binary logistic regression.

**Results**. Over three-quarters (79%) of obstetric care providers had adequate knowledge of nonpharmacological pain relief methods. The majority (85.5%) of the participants had a positive attitude toward NPPR in labour pain management, with the mean scores ranging from 3.55–4.23 for all sub-items. Obstetric care providers acknowledged that patient belief, lack of time, and workload were the strongest barriers to offering nonpharmacological pain relief methods for labour pain 67.6%, 64.5%, and 61.3%, respectively. In binary logistic regression analysis, the in-service training related to nonpharmacological pain relief (AOR = 5.871 (2.174–15.857), $p = 0.000$), (AOR = 3.942 (1.926–11.380), $p = 0.013$) and years of work experience (AOR = 1.678 (1.080–2.564), $p = 0.019$), (AOR = 1.740 (1.188–2.548), $p = 0.003$) were significantly associated with obstetric care providers' knowledge and attitudes regarding nonpharmacological pain relief ($p \leq 0.05$).

**Conclusion**. Although most OPCs have adequate knowledge and a positive attitude regarding NPPR, they need motivational strategies to enhance their utilization. In addition, an effort should be made to decrease OPCs' workload to provide more time for NPPR application and patient education. Training courses and in-service training can

play an important role in enhancing NPPR knowledge and attitude and, consequently, its application. Also, in each working unit, the policymakers should provide clear guidelines and policies that enhance and control the utilization of NPPR.

## INTRODUCTION

Pain is defined by the International Association for the Study of Pain as an unpleasant sensory and emotional experience associated with actual or potential tissue damage (*Raja et al., 2020*). Labour pain is a complex human experience and is greatly affected by numerous factors which make it a unique experience for each woman. However, labour pain is rated as severe by the majority of women; 90% of them reported satisfaction with the experience three months postpartum. This may be due to the positive labour outcomes and the effective pain management during labour (*Labor & Maguire, 2008*). The anatomical and physiological explanation of labour pain illustrated that it has two main components, visceral and somatic, and the process of cervical dilation has a contributing role in the two components. Visceral pain starts in the early first stage and continues during the second stage of labour due to pressure created by the uterine contraction on the cervix and lower uterine segment, leading to stretching and distension and activating excitatory nociceptive afferents. Alongside the visceral pain, somatic pain occurs in the late first and second stages of labour. Somatic pain results from the severe stretching and ischemia generated by fetal descent in the cervix, vaginal, perineum, and pelvic floor (*Gonzalez et al., 2016*; *Labor & Maguire, 2008*).

Numerous physical and psychological factors can contribute to the severity of labour pain. Physical factors include frequency, duration, and intensity of contraction. Psychological factors include stress, anxiety, and fear (*Siyoum & Mekonnen, 2019*). Inadequately controlled labor pain leads to negative or upsetting childbirth experiences. labour pain management is critical to improve the birth experience and decrease the incidence of postpartum depression (*Mo et al., 2022*).

Nonpharmacological pain relief (NPPR) methods can be utilized to reduce pain, alleviate suffering and enhance women's well-being during labour (*Heim & Makuch, 2022*). NPPR methods are recommended by the World Health Organization (WHO), among other sources of pain relief, to provide a positive birth experience (*WHO, 2018*). These methods are safe for both mother and fetus, have no side effects, do not affect labour progress, are cost-effective, and delay the use of pharmacological pain relief. NPPR helps women tolerate pain and have a more positive birth experience (*Smith et al., 2018*; *Boaviagem et al., 2017*). Furthermore, NPPR reduces negative outcomes associated with pharmacological pain relief methods and improves obstetric outcomes (*Gallo et al., 2018*). Many women prefer NPPR methods such as music and massage therapy, heat applications, deep breathing exercises,

position change, aromatherapies, acupressure, relaxation, acupunctures, transcutaneous electrical nerve stimulation (TENS), and hydrotherapy (*Adams et al., 2015*; *Boaviagem et al., 2017*; *Benfield, Heitkemper & Newton, 2018*).

Research indicates that the majority of women report being able to manage labor pain using NPPR methods and reported high satisfaction with this approach (*Czech et al., 2018*). Therefore, obstetric care providers (OCPs) play a crucial role in managing pain, promoting patient comfort, and aiding in the recovery of patients during their hospital stay. However, studies have shown that many hospitalized patients do not receive nonpharmacological interventions for pain relief (*Rantala, Hakala & Pölkki, 2022*), which can negatively affect their physical, emotional, and spiritual well-being as well as increase postpartum complications and healthcare costs (*Karabulut, Gurcayir & Aktas, 2016*).

The knowledge and attitude of nurses and other OCPs greatly influence the utilization of NPPR methods. Unfortunately, evidence has found that approximately two-thirds of OCPs have inadequate knowledge (66.9%) and unfavorable attitudes (65.5%) toward NPPR (*Kheshti et al., 2016*; *Eyeberu et al., 2022*), resulting in the underutilization of these methods (30.4%) (*Bishaw, Sendo & Abebe, 2020*) Several barriers prevent OCPs from implementing NPPR methods (*Bradfield et al., 2019*). Many OCPs doubt the effectiveness of NPPR compared to pharmacological options (*Boateng, Kumi & Diji, 2019*). Moreover, OCPs view NPPR methods as time-consuming and impractical due to their heavy workload, inadequate staffing, and limited clinical time (*Klomp et al., 2016*). In addition to limited knowledge, negative attitudes toward nonpharmacological pain relief and choice of obstetric care providers and patients can negatively affect the NPPR application. Among the aforementioned barriers, the knowledge and attitude of the obstetric care providers are the most vital (*Bishaw, Sendo & Abebe, 2020*). Literature suggests further research on the barriers that prevent obstetric care providers from offering nonpharmacological pain relief in maternity care (*Boateng, Kumi & Diji, 2019*). Nonpharmacological pain relief is considered suitable to make labor pain more controllable and tolerable. The first and most important step in nonpharmacological pain relief application is to evaluate the current situation. In Saudi Arabia, there are scarce studies, and there is a gap between the obstetric care provider's knowledge and attitude regarding nonpharmacological labour pain relief and its application in a real situation; therefore, it is very important to explore the barriers to nonpharmacological labour pain relief application which varies by culture, policies, and places. Due to this lack of information, the current study aims to evaluate nonpharmacological pain relief-related knowledge, attitudes, and barriers among obstetric care providers in Najran, Saudi Arabia.

## Research questions

- What is the level of obstetric care providers' knowledge and attitude regarding nonpharmacological pain relief?
- What are the barriers to nonpharmacological pain relief utilization by obstetric care providers?
- What are the demographic and work-related predictors of nonpharmacological pain relief knowledge and attitudes among obstetric care providers?

## MATERIALS & METHODS

### Study design and participants

A cross-sectional analytical study was performed at maternity departments where labour is expected to occur (delivery room, emergency department, and inpatient maternity departments) at Maternal and Children Hospital (MCH), Najran, KSA. Najran City is the administrative capital of Najran Province. It is located in southwest Saudi Arabia and has one large specialized hospital for maternity and children, serving about 595,705 people. A convenience sample of OCPs (nurses, midwives, and physicians) working in the previously mentioned departments in MCH and providing written informed consent was included in the study. Obstetric care providers with less than one year of work experience in the hospital were excluded from the study.

### Sampling

The Epi Info free sample size calculator (https://www.openepi.com/SampleSize/SSCohort.htm) was used to calculate the sample size. The total number of obstetric care providers working in the delivery room, emergency department, and inpatient maternity departments was 245, according to the data obtained from the MCH administration. The parameters used for sample size calculation were 99.9% CI, 5% margin error, and a power of 99%; the prevalence of adequate NPPR-related knowledge was 54.2% % from the prior study (*Bishaw, Sendo & Abebe, 2020*). The calculated sample size was 179, and we added 10% for the estimated nonresponse rate and the incomplete information. Thus, the required sample size was 197. In case of the selected OCPs refused participation, they were replaced by another. The self-reported questionnaire was distributed to all OCPs ($n = 245$), and 205 questionnaires were pooled. Then, 19 questionnaires were excluded due to incomplete and inconsistent information, so 186 questionnaires were analyzed. The participants in the current study were 19 physicians, 144 nurses, and 23 midwives (Fig. 1).

### Data collection

The researchers developed a structured self-reported questionnaire based on recent similar studies (*Jira et al., 2020*; *Mohamed Bayoumi, Khonji & Gabr, 2021*). It was prepared in English and involved five main sections: demographic data, work-related data, NPPR-related knowledge quiz, the attitude scale toward NPPR, and the perceived barriers scale to offer NPPR.

**The demographic data section** comprised age, religion, sex, nationality, marital status, educational level, and monthly income.

**Work-related data** included profession, years of work experience, provider-patient ratio, working hours, availability of NPPR guidelines, and training related to NPPR.

**The knowledge section** was developed to evaluate the NPPR definition, main types, benefits, and physiological background. The scale is composed of eight dichotomous and multiple choice questions scored as the correct answer (2), incomplete answer (1), and incorrect answer (0), inadequate knowledge considered at less than 60% (0–9.5), and adequate knowledge at 60% and more (9.6–16).

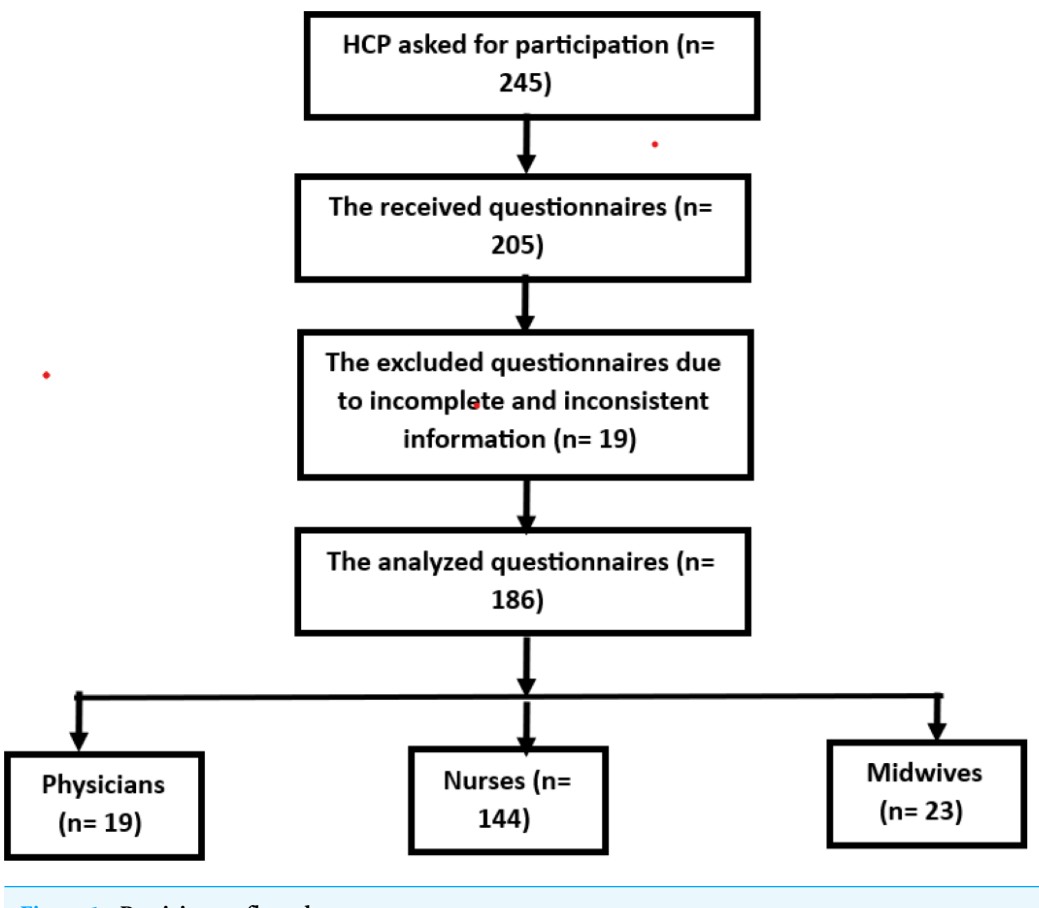

**Figure 1  Participants flow chart.**

**The attitude section:** The scale comprised ten items to assess the OCPs' attitude toward NPPR rated on a 5-point Likert scale ranging from strongly agree (5) to strongly disagree (1). The overall scale score ranged from 10–50; the participants were considered to have a negative attitude if their overall score fell between 10–30 and positive if their overall score fell between 31–50.

**The perceived barriers section**: The scale was developed to assess the perceived barriers to offering NPPR methods in labor pain management. It comprised 12 statements rated on a 5-point Likert scale ranging from strongly agree (5) to strongly disagree (1). The participants were considered to have barriers if they agreed and strongly agreed with the response.

## Instrument validity and reliability

The researchers developed the questionnaire; then, it was tested for face, content, and construct validity by an expert panel of 4 professors of obstetric care and a biostatistician. The instrument's reliability was assessed by Cronbach's Alpha test. The test results of attitudes, knowledge, and perceived barriers sections were 0.77, 0.78, and 0.81, respectively.

## Data collection procedures

Data collection started from April 1 to May 26 2023. The researchers disseminated the self-reported questionnaires in paper form to OCPs. To improve accessibility and collaboration among OCPs, one of them was selected as a data collector. The data collector was briefed on the research proposal, data collection instrument, and ethical considerations before beginning data collection.

Two of the researchers assisted in data collection; they have previous experience in data collection. The other data collector was a bachelor's degree holder with previous experience in data collection. Before data collection, two meetings were provided to the data collector regarding the study proposal, interview schedule, and research ethics.

## Ethical considerations

Ethical approval was obtained in four steps: (1) approval from the deanship of scientific research at Najran University (NU/DRP/MRC/12/2). (2) Approval from the ethical committee at Najran Health Affairs (IRB: 2023-06E), (3) permission from the hospital administration to begin data collection, and (4) obtaining written informed consent from participants. Participants were informed about their right to decline participation without any consequences, and all data gathered was kept confidential and utilized for research purposes only.

## Data quality control

Two meetings were held between the principal investigator and the data collectors to explain how to communicate with the study participants, the interview schedule, the study proposal, and research ethics. At the end of each data collection day, the collected questionnaires were examined and reviewed for missing information. Data collectors confirmed the clarity, completeness, and consistency of handwriting and the absence of any errors or ambiguities. Any problems raised during data collection were discussed, and appropriate solutions were provided. A pilot study was conducted with 16 healthcare providers before actual data collection to determine the accuracy of responses, clarity of language, and appropriateness of tools. The necessary changes were made to the tool based on the pilot study results. The modified tool was then used to collect data from all participants.

## Statistical analyses

The data were entered into SPSS version 23, and the necessary analysis was done. The data was analyzed using various methods such as number and percentage for categorical variables and mean and standard deviation for numerical variables. The overall knowledge and attitude were calculated by summing items. NPPR-related knowledge was categorized into adequate (coded 1) and inadequate (coded 0). The attitude toward NPPR was categorized into positive (coded 1) and negative (coded 0). Bivariate and multivariate analyses were done to determine the factors associated with NPPR-related knowledge and attitude variables utilizing binary logistic regression. All variables with $P \leq 0.25$ in the bivariate analysis were included in the final multivariate analysis model to handle all potential confounding factors. The Hosmer-Lemeshow statistic and Omnibus test were

used to test the model goodness of fit. The model was considered a good fit since it was found to be insignificant for the Hosmer-Lemeshow statistic ($p = 0.410$) and significant for Omnibus tests ($p = 0.001$). The multi-co-linearity test assessed the correlation between demographic and work-related variables *via* variance inflation factor and standard error; no variables were observed with variance inflation factor of $> 10$ and standard error $> 2$. The direction and power of statistical association were evaluated by the odds ratio with 95% confidence interval (CI). The adjusted odds ratio (AOR) along with 95% CI was estimated to determine the factors associated with NPPR-related knowledge and attitude using multivariate analysis in the binary logistic regression. The significant level of association was considered at $p < 0.05$.

## RESULTS

### Participants' demographic variables

Of all study participants, 174 (93.5%) were females, with a mean age of $37.25 \pm 8.71$ years. Approximately half of them (48.4%) were Indian, and 47.8% were Christian. Regarding marital and educational status, 72.0% were married, and 70.4% had Bachelor's degrees. About half (48.9%) of the participants had enough monthly income (Table 1).

### Work-related factors to NPPR among OCPs

The majority (77.4%) of the participants were nurses; an equal proportion (56.5%) reported an undetermined provider-patient ratio and worked eight hours per day. All (100.0%) participants reported not having the NPPR guideline in MCH. More than half (57.5%) received training related to NPPR during their formal education, and only 12.9% received training sessions after employment (Table 2).

### NPPR-related knowledge among OCPs

Over three-quarters (79.0%) of OCPs had adequate total knowledge about NPPR. Among the participants, 87.1% knew the correct definition of NPPR, and 78.0% were aware of the NPPR Benefits. Regarding the NPPR types, the majority of them were aware of the different types, such as co-cognitive-behavioral, physical, emotional, environmental comfort, and patient-family involvement 79.6%, 86.6%, 82.2%, 83.8%, and 78.5%, respectively (Table 3).

### OCPs' attitudes toward NPPR

OCPs' attitudes toward NPPR are illustrated in Table 4. The majority (85.5%) of the participants had a positive attitude toward NPPR in labour pain management, with the mean scores ranging from 3.55–4.23 for all scale items. The highest mean score was about the belief that they had a responsibility and obligation to manage pain ($4.23 \pm 0.70$); NPPR methods have lower side effects than medication ($4.15 \pm 0.80$) and can be used at home ($4.22 \pm 0.64$) (Table 4).

### Barriers to offering NPPR methods among OCPs

OCPs acknowledged that patient belief, lack of time, and workload were the strongest barriers to offering NPPR methods in labour pain management 67.7%, 64.5%, and 61.3%,

**Table 1  Participants' demographic variables (n = 186).**

| Demographic data | No | % |
|---|---|---|
| **Sex** | | |
| − Male | 12 | 6.5 |
| − Female | 174 | 93.5 |
| **Age in years (mean ±SD)** | 37.25 ± 8.71 | |
| **Nationality** | | |
| − Saudi | 21 | 11.3 |
| − Egyptian | 17 | 9.1 |
| − Sudanese | 3 | 1.6 |
| − Filipino | 55 | 29.6 |
| − Indian | 90 | 48.4 |
| **Religion** | | |
| − Muslim | 65 | 34.9 |
| − Christian | 89 | 47.8 |
| − Hindu religion | 25 | 13.4 |
| − Others | 7 | 3.8 |
| **Marital status** | | |
| − Single | 48 | 25.8 |
| − Married | 134 | 72.0 |
| − Divorced | 2 | 1.1 |
| − Widowed | 2 | 1.1 |
| **Educational level** | | |
| − High diploma | 41 | 22.0 |
| − Bachelor's degree | 131 | 70.4 |
| − Master's degree | 14 | 7.5 |
| **Monthly income** | | |
| − Not enough | 79 | 42.5 |
| − Enough | 91 | 48.9 |
| − Enough and can save | 16 | 8.6 |

respectively. At the same time, the lowest barriers related to insufficient motivation (6.5%) and lack of equipment (16.1%) (Fig. 2).

## Demographic and work-related predictors of NPPR knowledge and attitude among OCPs

In binary logistic regression analysis, the training related to NPPR and years of work experience were significantly associated with OCPs' knowledge and attitudes. However, educational level was found to be associated only with NPPR-related knowledge. A Master's degree qualification (AOR =3.353 (1.196–11.335) $p = 0.043$) increased the probability of having adequate knowledge by 3.3 times compared with a high diploma. Moreover, those participants who participated in in-services training regarding NPPR were more likely to have adequate knowledge and positive attitudes than those who didn't participate (AOR = 5.871 (2.174–15.857) $p = 0.000$) and (AOR = 3.942 (1.926–11.380) $p = 0.013$), respectively. In addition, one year increase in work experiences increased the OCPs'

**Table 2** Work-related factors to nonpharmacological pain relief among obstetric care providers (*n* = 186).

| Work-related factors | No | % |
|---|---|---|
| **Profession** | | |
| — Physician | 19 | 10.2 |
| — Nurse | 144 | 77.4 |
| — Midwife | 23 | 12.4 |
| **Providers: patient ratio** | | |
| — 1:4 | 33 | 17.7 |
| — 1: 6 | 10 | 5.4 |
| — 1: 8 | 38 | 20.4 |
| — Undetermined | 105 | 56.5 |
| **Working hours** | | |
| — 8 | 105 | 56.5 |
| — 12 | 62 | 33.3 |
| — More than 12 | 19 | 10.2 |
| **Availability of guidelines for using nonpharmacological pain relief in the unit.** | | |
| — Yes | 0 | 0.0 |
| — No | 186 | 100.0 |
| **Training related to nonpharmacological pain relief** | | |
| — Never received | 41 | 22.0 |
| — Yes, during my formal education | 107 | 57.5 |
| — yes, during my postgraduate education | 14 | 7.5 |
| — Yes, training session after employment. | 24 | 12.9 |
| **Years of experience (mean ±SD)** | 10.71 ± 6.59 | |

**Table 3** Nonpharmacological pain relief -related knowledge among obstetric care providers (*n* = 186).

| Nonpharmacological pain relief -related knowledge | Correct answer | |
|---|---|---|
| | No | % |
| **Definition of nonpharmacological pain relief.** | 162 | 87.1 |
| **The main types of nonpharmacological pain relief** | | |
| — Co-cognitive-behavioral | 148 | 79.6 |
| — Physical | 161 | 86.6 |
| — Emotional | 153 | 82.2 |
| — Environmental comfort | 156 | 83.8 |
| — Patient-family involvement | 146 | 78.5 |
| **Benefits of nonpharmacological pain relief.** | 145 | 78.0 |
| **Nonpharmacological pain relief methods have a physiological background in the body.** | 147 | 79.0 |
| **Total knowledge** | | |
| — Inadequate | 39 | 21.0 |
| — Adequate | 147 | 79.0 |

**Table 4  Obstetric care providers' Attitudes toward nonpharmacological pain relief (n = 186).**

| Statement | Mean | SD |
|---|---|---|
| **I think that nonpharmacological pain relief methods are** | | |
| Have lower side effects than medication | 4.15 | 0.80 |
| Lower cost | 3.76 | 0.88 |
| More available | 3.65 | 0.80 |
| Patient-centered | 3.55 | 0.73 |
| Building trust in the therapeutic relationship | 4.03 | 0.68 |
| It can be used at home. | 4.22 | 0.64 |
| More relaxing | 3.84 | 0.90 |
| More available | 4.08 | 0.73 |
| Necessary for managing pain | 4.15 | 0.80 |
| The belief that you have a responsibility and obligation to manage Pain | 4.23 | 0.70 |
| **Total attitudes score** | **No** | **%** |
| — Negative | 27 | 14.5 |
| — Positive | 159 | 85.5 |

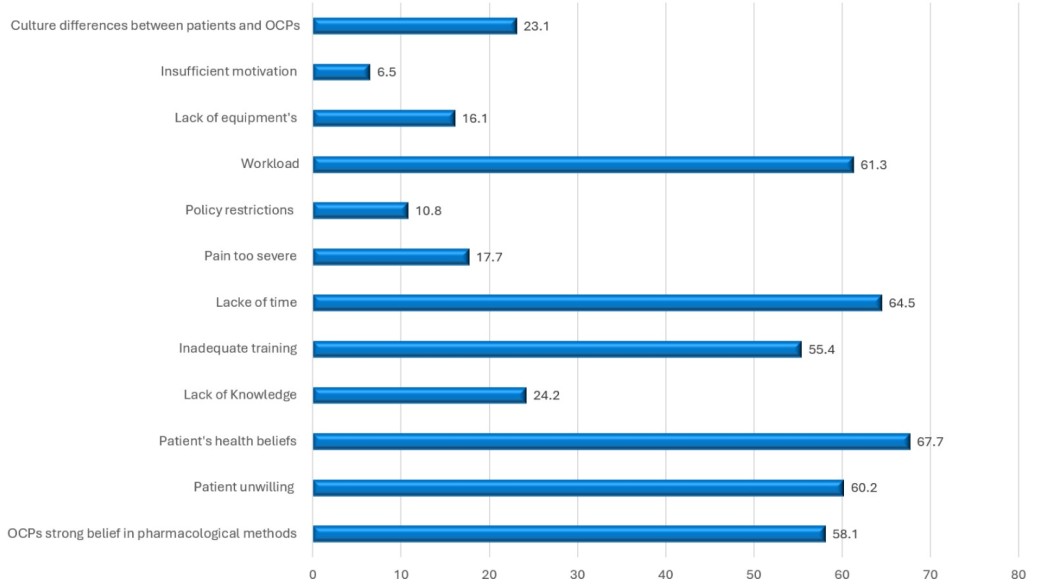

**Figure 2  Barriers to offering nonpharmacological pain relief methods among obstetric care provides.**
Note: The total is not mutually exclusive.

probability of having adequate knowledge and positive attitudes by 1.7 times (AOR = 1.678 (1.080–2.564), $p = 0.019$) and (AOR = 1.740 (1.188–2.548), $p = 0.003$), respectively (Table 5).

**Table 5  Demographic and work-related predictors of nonpharmacological pain relief knowledge and attitude among obstetric care providers.**

| Demographic/ work-related predictors | Knowledge | | Attitude | |
|---|---|---|---|---|
| | AOR (95% CI) | *p* | AOR (95% CI) | *P* |
| **Sex** | | | | |
| — Male | Ref | | Ref | |
| — Female | 1.382 [0.284–6.726] | 0.689 | 1.315 [0.342–4.913] | |
| **Nationality** | | | | 0.534 |
| — Saudi | Ref | | Ref | |
| — Egyptian | 2.492 [0.656–9.473] | 0.180 | 0.618 [0.336–1.060] | 0.084 |
| — Sudanese | 1.305 [0.348–4.900] | 0.693 | 0.959 [0.398–2.313] | 0.919 |
| — Filipino | 1.059 [0.292–3.834] | 0.931 | 0.921 [0.516–1.658] | 0.817 |
| — Indian | 1.882 [0.527–6.718] | 0.330 | 1.530 [0.570–4.081] | 0.377 |
| **Religion** | | 0.336 | | 0.792 |
| — Muslim | Ref | | Ref | |
| — Cristian | 0.858 [0.663–1.165] | 0.342 | 1.034 [0.677–1.645] | 0.861 |
| — Hindu religion | 0.481 [0.100–2.321] | 0.362 | 1.070 [0.625–1.804] | 0.779 |
| — Others | 0.393 [0.038–4.11] | 0.436 | 1.063 [0.727–1.535] | 0.783 |
| **Marital status** | | | | 0.756 |
| — Single | Ref | 0.340 | Ref | |
| — Married | 0.479 [0.160–1.431] | 0.187 | 1.010 [0.989–1.052] | 0.340 |
| — Divorced | 0.704 [0.247–2.002] | 0.510 | 0.987 [0.889–1.116] | 0.948 |
| — Widowed | 0.718 [0.249–2.070] | 0.540 | 0.920 [0.775–1.065] | 0.200 |
| **Educational level** | | 0.017* | | 0.328 |
| — High diploma | Ref | | Ref | |
| — Bachelor's degree | 1.234 [0.857–3.334] | 0.557 | 0.954 [0.912–1.019] | 0.102 |
| — Master's degree | 3.353 [1.196–11.335] | 0.043* | 0.827 [0.416–1.738] | 0.621 |
| **Monthly income** | | 0.643 | | 0.685 |
| — Not enough | Ref | | Ref | |
| — Enough | 0.857 [0.348–2.114] | 0.738 | 0.835 [0.274–2.566] | 0.771 |
| — Enough and can save | 0.797 [0.333–1.910] | 0.611 | 3.637 [0.146–85.770] | 0.411 |
| **Age in years** | 0.963 [0.90–1.029] | 0.263 | 0.975 [0.913–1.065] | 0.724 |
| **Profession** | | 0.556 | | 0.431 |
| — Physician | Ref | | Ref | |
| — Nurse | 0.302 [0.021–4.327] | 0.378 | 1.305 [0.776–2.524] | 0.242 |
| — Midwife | 0.504 [0.025–10.354] | 0.657 | 1.328 [0.515–3.307] | 0.532 |
| **Providers: patient ratio** | | 0.831 | | 0.375 |
| — 1:4 | Ref | | Ref | |
| — 1: 6 | 0.663 [0.102–4.321] | 0.667 | 0.881 [0.653–1.187] | 0.434 |
| — 1: 8 | 1.549 [0.369–6.495] | 0.550 | 1.270 [0.665–2.339] | 0.458 |
| — Undetermined | 1.239 [0.400–3.841] | 0.710 | 0.873 [0.645–1.189] | 0.438 |

*(continued on next page)*

**Table 5** (*continued*)

| Demographic/ work-related predictors | Knowledge | | Attitude | |
|---|---|---|---|---|
| | AOR (95% CI) | *p* | AOR (95% CI) | *P* |
| **Working hours** | | 0.921 | | 0.685 |
| − 8 | Ref | | Ref | |
| − 12 | 1.152 [0.424–3.130] | 0.781 | 0.845 [0.274–2.566] | 0.771 |
| − More than 12 | 0.851 [0.208–3.492] | 0.823 | 3.647 [0.146–85.770] | 0.410 |
| **Training related to nonpharmacological pain relief** | | | | 0.014[*] |
| − Never received | Ref | 0.004[*] | Ref | |
| − Yes, during my formal education | 5.750 [0.658–50.235] | 0.114 | 2.864 [1.231–6.643] | 0.024[*] |
| − yes, during my postgraduate education | 5.333 [0.638–44.579] | 0.122 | 2.458 [1.098–6.117] | 0.040[*] |
| − Yes, in-service training sessions. | 5.871 [2.174–15.857] | 0.000[*] | 3.942 [1.926–11.380] | 0.013[*] |
| **Years of experience** | 1.678 [1.080–2.564] | 0.019[*] | 1.740 [1.188–2.548] | 0.003[*] |

**Notes.**
AOR, Adjusted Odd Ratio; CI, Confidence Interval.
[*] significant at $p < 0.05$.

## DISCUSSION

Labor is a unique experience where contradictory emotions are present. Pain is an inseparable part of the labor process, and it should probably be managed without side effects for the mother, the progress of labor, and the infant. NPPR is considered suitable to make labor pain more controllable and tolerable. The first and most important step in NPPR application is to evaluate the current situation. In Saudi Arabia, there are no available studies in the international database to evaluate the knowledge, attitude, and barriers to NPPR application for labor pain; therefore, it's the first Saudi study performed for this aim. In the present study, over three-quarters of the OCPs had adequate knowledge about NPPR definition, benefits, main types, and physiological background.

In the same line, *Eyeberu et al. (2022)* found that 82.7% of the OCPs had adequate knowledge regarding NPPR, and only 12.5% only knew all types of NPPR. The most known NPPR type among their participants was psychotherapy and massaging (*Eyeberu et al., 2022*). Besides, (*Emelonye et al. (2017)* illustrated that most of the midwives who participated in the study acknowledged the husband's presence and support during labor as an important NPPR method, but only one-quarter of them applied it in real practice. In addition, *Boateng, Kumi & Diji (2019)* explored the midwives and nurses' experience of NPPR utilization in a qualitative study. They illustrated that the majority of their participants have good knowledge about NPPR but demonstrated low knowledge regarding many types of it. Furthermore, *Bishaw, Sendo & Abebe (2020)* reported that 54.2% of their OCPs had satisfactory knowledge regarding NPPR methods. The participants reported that psychotherapy, ambulation, massage, patient education, and allow companionship were the most popular and known nonpharmacological pain methods. In addition, *Jira et al. (2020)* investigated the nurses' knowledge and attitude regarding NPPR and its associated factors. They found that more than half of their participants had adequate knowledge regarding NPPR benefits, while 38.3% did not know its types. The possible explanation for the satisfactory knowledge about NPPR among the OCPs is that the majority of them

had Bachelor's degrees or postgraduate education. In addition, the minimum educational requirement to work as a health care provider all over the world is a high diploma. Therefore, it is expected that most OCPs will have satisfactory knowledge regarding different aspects of patient care, including NPPR. Even if the OCP had limited information about any aspect of patient care, he could access it smoothly through the open database.

On the other hand, a recent Iranian study found that 73.6% of their healthcare providers had limited knowledge regarding complementary and alternative therapy modalities (*Jafari et al., 2021*). The differences between the current study and the Iranian one related to knowledge score may be related to the type of knowledge evaluated. The current study evaluated knowledge regarding definition, modalities, benefits, and physiological background, while the Iranian one evaluated only complementary and alternative therapy modalities.

The present study showed that most of the participants had a positive attitude toward NPPR methods in labour pain management, with the mean scores ranging from 3.55-4.23 for all items. Labour and childbirth are considered normal physiologic processes by OCPs; therefore, a large proportion of them thought that the use of pharmacological pain relief methods was unnecessary. However, 87.6% of them perceive labour pain as severe and should be managed properly to enhance a positive birth experience without using pharmacological pain relief methods, which may delay labor and cause fetal distress. Consequently, a positive attitude toward NPPR is common among midwives, obstetricians, and nurses (*Bishaw, Sendo & Abebe, 2020*). Furthermore, an Egyptian study illustrated that 69.0% to 89.7% of the OCPs had a positive attitude toward NPPR utilization during the first stage of labor but reported little benefit from it during the second stage (*Mousa et al., 2018*). A recent Iranian study found that 79% of healthcare providers had a positive attitude toward the utilization of NPPR, and they thought that both mind and body should be managed equally and in a synchronized manner (*Jafari et al., 2021*). Besides, *Jira et al. (2020)* reported that around half of the maternity nurses in their study have a positive attitude toward NPPR regardless of their ability to apply it in clinical practice. They further added that NPPR is very effective for mild to moderate pain and has little effect on severe pain.

On the contrary, *Eyeberu et al. (2022)* studied obstetricians' utilization and attitude toward NPPR for Ethiopian women. They illustrated that although a high percentage of their participant utilized NPPR methods, 65.5% of them had an unfavorable attitude toward it. Disparities between the Ethiopian study and the current one may be due to participant sex, where 43.1% of their participants were male compared to only 10.5% in the current study

Concerning barriers to offering NPPR methods, OCPs acknowledged that patient belief, lack of time, and workload were the strongest barriers to providing NPPR methods during labour. At the same time, the lowest barriers are related to insufficient motivation and lack of equipment. Most NPPR methods require adequate training, time, and relaxation from the healthcare providers; therefore, the current study reported a lack of time and high workload as the most significant barriers to NPPR utilization. Along the same line, an Ethiopian study reported that lack of adequate training, high patient flow, and high workload were the most

important barriers to NPPR utilization (*Bishaw, Sendo & Abebe, 2020*). Furthermore, the nurse should have a strong belief and commitment to NPPR to overcome other obstacles to its implementation. The qualitative study conducted by *Boateng, Kumi & Diji (2019)* reported that the most important barrier among their participants was the strong belief in pharmacological pain relief compared with NPPR. They elaborated that most midwives reported that NPPR might induce relaxation, but it does not take the pain away. In fact, pain is an inseparable part of the normal labor experience because of physiological reasons, and it has an important role in the labor process; therefore, it needs to be tolerable and controllable but rarely relived using NPPR. NPPR can delay the use of pharmacological analgesia, decreasing the dose required and consequently decreasing the expected side effects (*Gallo et al., 2018*; *Bonapace et al., 2018*). *Mwakawanga et al. (2022)* reported that the limited number of healthcare providers and high workload discourage them from applying NPPR or continuing its utilization, especially for some methods that require the continuous presence of the OCPs. In the same line with the current study, the lack of facilities to apply some NPPR methods and client beliefs regarding it were also reported barriers (*Bonapace et al., 2018*). Other studies reported that many women were not ready to utilize NPPR and preferred pharmacological methods (*Anarado et al., 2015*; *Thomson et al., 2019*). In addition, *Mousa et al. (2018)* reported that the most common barriers among their participants were hospital-related factors such as lack of facilities, policies and guidelines, and high workload. They further added that clinician-related factors, such as their knowledge and attitude toward NPPR, were important barriers to its utilization.

In binary logistic regression analysis, the training related to NPPR and years of work experience were significantly associated with OCPs' knowledge and attitudes regarding NPPR. However, educational level was found to be associated only with knowledge. Moreover, those participants who participated in in-service training regarding NPPR were more likely to have adequate knowledge and positive attitudes than those who didn't participate. In addition, a one-year increase in work experiences increased the OCPs' probability of having adequate knowledge and positive attitudes by 1.7 times.

Most undergraduate health education programs did not give much attention to complementary and alternative medicine. However, higher education may contain complete courses related to complementary medicine, including NPPR methods. The current study showed that a master's degree qualification increased the probability of having adequate knowledge by 3.3 times compared with a high diploma. *Bishaw, Sendo & Abebe (2020)* found that higher education increased the care provider's probability of practicing NPPR by 3.45 times. Besides, *Jira et al. (2020)* reported that nurses with postgraduate education reported a 12.2 times higher probability of having adequate knowledge regarding NPPR methods compared to diploma nurses. They further added that nurses with higher experience had a higher probability of having adequate knowledge when compared with nurses who reported less than one year of experience. They further added that nurses who received NPPR training had a 7.5 times higher probability of having higher knowledge and a 4.6 times higher probability of a positive attitude than nurses who never received training. The results of the current study highlighted that although most participants had good knowledge and positive attitudes regarding NPPR, they had

numerous barriers related to its application in the practical setting. These barriers need to be considered and solved to enhance NPPR application and, consequently, provide a more positive birth experience.

### Study strengths and limitations

This is the first study to evaluate OCPs' knowledge, attitude, and barriers to utilizing NPPR during labour in Saudi Arabia. This study can provide a database for future NPPR utilization strategies in Saudi hospitals. However, the desire for social acceptance and the nature of the interviewing schedule may result in subjective bias. Moreover, this study might not indicate a cause-effect relationship because of the cross-sectional design. A qualitative research approach may be better to address the barriers to NPPR utilization. Therefore, we recommended conducting further qualitative research to assess barriers to NPPR application in various healthcare settings.

## CONCLUSIONS

Most OCPs had adequate knowledge and a positive attitude regarding NPPR during labour. OCPs acknowledged that patient belief, lack of time, and workload were the strongest barriers to offering NPPR for labour pain management. At the same time, the lowest barriers are related to insufficient motivation and lack of equipment. Binary logistic regression showed that training related to NPPR and years of work experience were significantly associated with OCP's knowledge and attitudes regarding NPPR. However, educational level was associated only with NPPR knowledge.

### Recommendation and clinical implications

Although most of the OPCs have adequate knowledge and a positive attitude regarding NPPR, they need motivational strategies to enhance their utilization. In addition, an effort should be made to decrease OPCs' workload to provide more time for NPPR application and patient education. Training courses and in-service training can play an important role in enhancing NPPR knowledge and attitude and, consequently, its application. Besides, decision-makers should provide opportunities for healthcare providers to continue postgraduate education. Also, in each working unit, the policymakers should provide clear guidelines and policies that enhance and control the utilization of NPPR. Further research is recommended to use a qualitative approach to enrich the international database with more barriers regarding the utilization of NPPR.

## ACKNOWLEDGEMENTS

The authors are thankful to the participants who participated in the current study.

### Funding

The Deanship of Scientific Research at Najran University funded this work under the Distinguished Research Funding Program grant code (NU/DRP/MRC/12/2). The funders had no role in study design, data collection and analysis, decision to publish, or preparation of the manuscript.

### Grant Disclosures

The following grant information was disclosed by the authors:
The Deanship of Scientific Research at Najran University: NU/DRP/MRC/12/2.

### Competing Interests

The authors declare there are no competing interests.

### Author Contributions

- Heba Abdel-Fatah Ibrahim conceived and designed the experiments, performed the experiments, prepared figures and/or tables, authored or reviewed drafts of the article, and approved the final draft.
- Majed Said Alshahrani analyzed the data, prepared figures and/or tables, authored or reviewed drafts of the article, and approved the final draft.
- Amlak Jaber Al-Qinnah performed the experiments, prepared figures and/or tables, authored or reviewed drafts of the article, and approved the final draft.
- Wafaa Taha Elgzar conceived and designed the experiments, performed the experiments, prepared figures and/or tables, authored or reviewed drafts of the article, and approved the final draft.

### Human Ethics

The following information was supplied relating to ethical approvals (i.e., approving body and any reference numbers):
   The ethical committee at Najran health affairs

### Field Study Permissions

The following information was supplied relating to field study approvals (i.e., approving body and any reference numbers):
   ethical committee at Najran health affairs (IRB: 2023-06E)

### Data Availability

   The raw data are available in the Supplemental File.

### Supplemental Information

Supplemental information for this article can be found online at http://dx.doi.org/10.7717/peerj.16862#supplemental-information.

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
