# Peer review of "Nonpharmacological pain relief for labour pain: knowledge, attitude, and barriers among obstetric care providers"

_PeerJ, doi:10.7717/peerj.16862_

## Round 0.1 · original submission · Major Revisions

There are several important aspects of your submitted manuscript that have been identified during the peer review process that require your attention and review. These factors must be properly addressed before your manuscript is further considered for publication.

Reviewers have noted several issues with format and style. We encourage you to review these areas carefully to ensure they meet publishing standards.

In addition, you need a clear and compelling rationale for your study, especially from a quantitative research perspective. It is important to clarify why this study is important and how it contributes to existing knowledge. This will not only strengthen your manuscript but also clarify its importance in the broader academic discourse.

In terms of methodology, an important aspect to focus on is the sense of controlling the assumptions in your logistic regression models, especially justification for the control of assumptions in your logistic regression models, specifically concerning multicollinearity. It is therefore important to demonstrate how well these concepts were addressed in your study.

In addition, the chart shows the consistency between the p-values ​​and the 95% confidence intervals of odds ratios for some variables. This is of particular concern, as it directly affects the consistency and reliability of your results. Better clarification and correction of these differences are imperative.

Further development is required in the limitations section of your manuscript to fully address the various limitations brought about by your research methodology. It is crucial to openly acknowledge these limitations in order to present a well-rounded and valid interpretation of your results.

In brief, there is a need for substantial revisions in areas such as adhering to reviewer feedback, formatting and writing style, justification for the study, rigor in methodology, consistency in statistical analysis, and thorough discussion of limitations. Your careful consideration and improvement in these aspects are vital for the progress of your manuscript in the publication process.

**Language Note:** The review process has identified that the English language must be improved. PeerJ can provide language editing services - please contact us at copyediting@peerj.com for pricing (be sure to provide your manuscript number and title). Alternatively, you should make your own arrangements to improve the language quality and provide details in your response letter. – PeerJ Staff

·

Basic reporting

There are a lot of errors in basic reporting format. I have provided a detail comments at the end.

Experimental design

The design is well explained.

Validity of the findings

It reported with standard of the journals. Details of my comments provided at the end.

Additional comments

Dear respected editor, thank you very much for the invitation to review this manuscript. The manuscript is well written. The labor pain management is neglected issue and it is important to be investigated. So, this can be an input for policy making. Even though, the manuscript is well written it have some shortcoming in abstract, introduction, methods and discussion sections.
Here below I provided specific comments.
1. Regarding the writing style, the use of font size is irregular. E.g., line number 18 and 19.
2. In the abstract, what makes your study different from other published paper. SO, the gap of the study should be explicitly stated.
3. On line number 19, rather than saying the end of May, please state the date explicitly.
4. On line number 24 and 25, the confidence of uncertainty should be reported.
5. On line number 28 and 29, each factors level of significance and effect size should be reported.
6. I found that your recommendation is not specific and strong. You should give specific and strong recommendations to the concerned bodies.
7. On the introduction section, paragraph one and two have different type of writing style and font use. Please be consistent.
8. In text citations are also different. Some are colored, some are highlighted, some have high font size. E.g., line number 39, 53, 54, 60……
9. The introduction should be supported with numbers. Such as the number of women get non-pharmacological labor pain management, proportion of OCP practiced NPLPM, proportion of OCPs who had favorable attitude, …
10. I found that the need to conduct this study is not well explored. If the case is barriers. It should be well discovered in the result section. It is crystal clear that qualitative approaches are better to address such kind of variables.
11. I found that the method section is well written. However, the writing style and grammar errors are a major challenge throughout the manuscript. Please be consistent thorough the paper and proof read the whole section.
12. It is better if you clearly show how, you measured the variable and how you maintain the quality of the data.
13. On the statistical analysis section, please clearly show how the model is fitted? How the assumptions are tested? How multi collinearity is checked? If you have done all those things, you need to report all findings.
14. On line number 203, the confidence interval contains 1, so it had no significant association b/n Master's degree qualification and knowledge. Ho you explain this?
15. The discussion section is well written. The major gap is it lacks interpretations of the some findings and implication of all findings.

·

Basic reporting

editing , formats, needed to some extend in all over of the manuscript
also i attached the comments inside the file

Experimental design

research questions not mentioned and research objectives
inclusion and exclusion criteria not present also

Validity of the findings

no comment

Additional comments

only revision and editing , also some information needed to be added as recommendations& implication of the study also many of references repeated in reference list
all my comments i attached and mentioned it in the attached file

---

## Round 0.2 · accepted · Accept

The authors have carefully responded to comments received from reviewers. The revised manuscript has made a significant leap from the original version. The revisions and supplements introduced based on the feedback from reviewers have improved not only clarity but also detail of research that is presented.

I am of the opinion that these changes have managed to satisfy all the concerns raised earlier hence making them acceptable for publishing. The revised version does not only meet the standards of PeerJ, but also adds value to its respective study discipline.

Given the significant progress made, I believe this manuscript is ready for publication.

·

Basic reporting

All comments are well addressed.

Experimental design

No comment

Validity of the findings

All comments are well addressed

Additional comments

All comments are well addressed.

·

Basic reporting

no comment

Experimental design

no comment

Validity of the findings

no comment

Additional comments

no comment and a valuable work, after the reviewer's comments well done